# De Novo Mosaic 6p23-p25.3 Tetrasomy Caused by a Small Supernumerary Marker Chromosome Presenting Trisomy Distal 6p Phenotype: A Case Report and Literature Review

**DOI:** 10.3390/diagnostics12102306

**Published:** 2022-09-24

**Authors:** Yu-Min Syu, Juine-Yih Ma, Tzu-Hsuen Ou, Chung-Lin Lee, Hsiang-Yu Lin, Shuan-Pei Lin, Chia-Jung Lee, Chih-Ping Chen

**Affiliations:** 1Department of Pediatrics, Far Eastern Memorial Hospital, New Taipei City 22021, Taiwan; 2Division of Genetics and Metabolism, Department of Pediatrics, MacKay Memorial Hospital, Taipei 10449, Taiwan; 3Department of Rare Disease Center, MacKay Memorial Hospital, Taipei 10449, Taiwan; 4Department of Medical Research, Division of Genetics and Metabolism, MacKay Memorial Hospital, Taipei 10449, Taiwan; 5Department of Childhood Care and Education, MacKay Junior College of Medicine, Nursing and Management, Taipei 11260, Taiwan; 6Department of Medicine, MacKay Medical College, New Taipei City 25245, Taiwan; 7Department of Medical Research, China Medical University Hospital, China Medical University, Taichung 40402, Taiwan; 8Department of Infant and Child Care, National Taipei University of Nursing and Health Sciences, Taipei 11219, Taiwan; 9Department of Obstetrics and Gynecology, Mackay Memorial Hospital, Taipei 10449, Taiwan; 10School of Chinese Medicine, College of Chinese Medicine, China Medical University, Taichung 40402, Taiwan; 11Institute of Clinical and Community Health Nursing, National Yang Ming Chiao Tung University, Taipei 11230, Taiwan; 12Department of Obstetrics and Gynecology, School of Medicine, National Yang Ming Chiao Tung University, Taipei 11230, Taiwan; 13Department of Medical Laboratory Science and Biotechnology, Asia University, Taichung 41346, Taiwan

**Keywords:** small supernumerary marker chromosomes, mosaic partial tetrasomy 6p, trisomy 6p, copy number variation sequencing, triplosensitivity

## Abstract

Small supernumerary marker chromosomes (sSMCs) derived from the chromosome 6 short arm are rare and their clinical significance remains unknown. No case with sSMC(6) without centromeric DNA has been reported. Partial trisomy and tetrasomy of distal 6p is a rare but clinically distinct syndrome. We report on a de novo mosaic sSMC causing partial tetrasomy for 6p23-p25.3 in a male infant with symptoms of being small for gestational age, microcephaly, facial dysmorphism, congenital eye defects, and multi-system malformation. Conventional cytogenetic analysis revealed a karyotype of 47,XY,+mar [25]/46,XY [22]. Array comparative genomic hybridization (aCGH) revealed mosaic tetrasomy of distal 6p. This is the first case of mosaic tetrasomy 6p23-p25.3 caused by an inverted duplicated neocentric sSMC with characteristic features of trisomy distal 6p. Comparison of phenotypes in cases with trisomy and tetrasomy of 6p23-p25.3 could facilitate a genotype–phenotype correlation and identification of candidate genes contributing to their presentation. The presentation of anterior segment dysgenesis and anomaly of the renal system suggest triplosensitivity of the *FOXC1* gene. In patients with microcephaly growth retardation, and malformation of the cardiac and renal systems, presentation of anterior segment dysgenesis might be indicative of chromosome 6p duplication, and aCGH evaluation should be performed for associated syndromic disease.

## 1. Introduction

A small supernumerary marker chromosome (sSMC) is a structurally abnormal additional chromosome that most often lacks a distinct banding pattern and is rarely identifiable by conventional banding cytogenetic analysis. The presence of an sSMC causes partial tris- or tetrasomy and the incidence rate is 0.44/1000 in newborn cases and approximately 0.75/1000 in prenatal testing, and only 30% of the cases are associated with clinical abnormalities [1]. Approximately 70% of SMC occur de novo and 30% of the cases are mosaic. Only some specific sSMC-related syndromes have been identified, whereas the genotype–phenotype correlations of most sSMC cases are yet to be established [2].

Partial tetrasomy of chromosome arm 6p is extremely rare. To date, only one case with partial tetrasomy of distal 6p has been reported in the literature, which was not associated with sSMCs [3]. In contrast, partial trisomy 6p is considered a clinically distinct syndrome reported in dozens of cases, most of which are associated with monosomy for segments of different chromosomes [4]. Pure duplication of distal 6p is rare with only 15 previous cases involving the duplication of 6p23-p25.3 [4,5,6,7,8,9,10,11,12,13,14]**;** characteristic phenotypes generally include low birth weight, developmental delay, growth retardation, craniofacial abnormalities, cataracts, congenital heart defects, glomerulopathy, and kidney and urinary tract anomalies.

We present a case of a male infant with de novo mosaic tetrasomy 6p23-p25.3 caused by a sSMC and compare the phenotype with previously reported cases with sSMC derived from chromosome 6p and cases of partial trisomy 6p. The genes reported to be responsible for the phenotype, namely *FOXC1* and *BMP6*, are discussed. The presentation of anterior segment dysgenesis and glomerulopathy might suggest triplosensitivity of the *FOXC1* gene.

## 2. Case Report

The proband is a male infant, the first child of non-consanguineous parents with a healthy 21-year-old mother and 23-year-old father. The mother had a previous miscarriage at 16 weeks of gestation. There was no family history of malformation or mental retardation. The mother had no history of tobacco or drug abuse. The mother received regular prenatal examinations without amniocentesis. The pregnancy was complicated with intrauterine growth restriction beginning at 30 weeks of gestation and oligohydramnios and a high umbilical artery systolic/diastolic ratio were noted two weeks before delivery. The child was born by vaginal delivery at 38 weeks gestation with a birth weight of 1615 g (<3rd centile), length of 46 cm (<3rd centile), and a fronto–occipital circumference of 30 cm (<3rd centile). Apgar scores were 7 at 1 min and 9 at 5 min. Immediately after birth, the patient required ventilatory support for respiratory distress and was hospitalized in the neonatal intensive care unit for two months. Physical examination by a genetics specialist revealed facial dysmorphism (Figure 1A,C) with depressed nasal bridge, bulbous nose with prominent columella, left low-set hypoplastic ear, auricular pit on right middle area on the crus of helix, and bilateral ear lobe crease. Hemangiomas were noted over bilateral upper eyelids, nose tip, and philtrum spreading to upper lip. He also presented with choanal atresia, bilateral corneal opacity (Figure 1B), hypospadias, and sacral dimple. There was no simian crease on the hands and feet but there were hypoplastic nails on the fourth and fifth fingers and toes. His survey for congenital infection, hematological and immunoglobulin profiles, and metabolic investigations were unremarkable.

His neurological examination revealed mild hyperspasticity, no hypotonia, and intact primitive reflexes, including sucking, rooting, and Moro reflex. He was fed via a nasogastric tube because of feeding difficulty. An electroencephalogram (EEG) revealed normal background activity and no epileptiform discharge. Neuroimaging studies of the brain and spine structures showed a small caliber for the left internal carotid artery with a suspicion of hypoplasia, no obvious myelination milestones, or brain or spinal structural abnormality.

His cardiovascular examination, including echocardiogram, showed a complete atrioventricular canal, small atrial septal defect, moderate valvular pulmonary stenosis with left pulmonary artery stenosis, and severe pulmonary hypertension; low-dose diuretic therapy with furosemide was given thereafter.

Evaluation of the genitourinary system revealed glandular type hypospadias, bilateral small kidneys with right moderate hydronephrosis and hydroureter, and left mild hydronephrosis. Voiding cystourethrography at two months of age showed bilateral high-grade vesicoureteral reflux. There were two recurrent episodes of urinary tract infections complicated with urosepsis, which were managed with intravenous antibiotic therapy during hospitalization.

His ophthalmological examination at four days of age revealed normal intraocular pressure of bilateral eyes, ring-shaped corneal opacities with suspected iridocorneal touch at the pupil margin, bilateral corectopia, and mydriasis. Anterior segment dysgenesis of bilateral eyes was diagnosed, which was indicative of Peter’s anomaly. He also failed a newborn hearing screen, and a follow-up evaluation indicated moderate to severe bilateral sensorineural hearing loss.

He had recurrent infections with pneumonia twice and urinary tract infections twice during the 3-month hospitalization, and all episodes were managed with intravenous antibiotic therapy. He was discharged at three months of age and followed up in the out-patient department. Global development was delayed with the absence of interaction with the environment. However, cardiomegaly and heart failure progressed during the follow-up period and he was hospitalized again at the age of four months. After a family meeting with multidisciplinary team, his parents decided to have him receive palliative care.

## 3. Results

Peripheral blood chromosome analysis was performed on the proband and GTG-banded chromosomes showed a supernumerary marker in approximately 53% of cells. The karyotype was described as 47,XY,+mar [25]/46,XY [22] (Figure 2A). Array comparative genomic hybridization studies (aCGH) were performed on a peripheral blood specimen using the CytoOneArray^®^ microarray (Phalanx Biotech Group, Taiwan). The origin of the supernumerary marker was determined to be a partial 6p duplication of 14.337 Mb at arr[GRCh37] 6p25.3p23 (211198_14548093) × 3–4 with a log2 ratio of 0.598 (Figure 2C,D), indicative of a mosaic partial tetrasomy of 6p. No other significant copy number changes were detected by aCGH. The repeated karyotypes of the proband at 3 months revealed 47,XY,+mar [26]/46,XY [14] (Figure 2B), indicating a decreased mosaicism level of cells containing sSMCs; SMC(6) had inverted duplicated shapes. The parents had normal karyotypes, indicating a de novo marker chromosome. The duplicated chromosomal section contained 65 OMIM genes, including *FOXC1* and *BMP6*.

## 4. Discussion

The poor clinical correlation of sSMCs remains a major problem in providing diagnosis and prognosis. In a previous study by Jang et al., approximately 40% of all SMCs were derived from non-acrocentric autosomes, with a risk of abnormal phenotypes of approximately 28% [15]. SMCs derived from chromosome 6 are rare. According to the sSMC database by Liehr [16], only nine cases were reported with clinical findings associated with sSMC purely derived from chromosome 6p, none of which involved purely distal 6p. The reported cases with clinically significant findings are summarized in Table 1 and Table 2, which had mosaicism of sSMC involving the pericentric region of 6p and centromere of chromosome 6, with the largest one involving 6p21.2/q12 [17]. The centromere was previously considered the fundamental structure that controls the segregation of chromosomes at meiosis and mitosis, until the discovery of the neocentromere in 1993 by Voullaire et al. [18]. Neocentromeres are ectopic centromeres that originate occasionally from non-centromeric regions of chromosomes and are capable of forming a primary constriction and assemble a functional kinetochore [19]. Neocentric SMCs are formed when an acentric chromosomal fragment is rescued by the formation of a neocentromere; they can be either an inverted duplication of the distal part of a chromosome arm resulting in an unbalanced rearrangement, or a balanced rearrangement into linear and circular SMCs after an interstitial deletion [20]. Regarding sSMC(6) not involving the centromere, there have been three cases with neocentromeres derived from chromosome 6 reported [17], two of which were ring chromosomes from 6q [16] and one from an inverted duplication of segments of distal 6q [20]. To date, there have been no reports of sSMC derived from purely interstitial distal 6p. Our patient had a mosaic sSMC(6) with an inverted duplicated shape, which was confirmed by aCGH as a mosaic tetrasomy of interstitial distal 6q. The inverted duplication of 6p23-6p25.3 and absence of centromere for chromosome 6 in the sSMC(6) suggest the formation of a neocentromere within the sSMC. Additionally, the initial mosaic level of 53%, is comparable with the theoretical ratio of mosaicism caused by formation of neocentromere via acentric fragment rejoining after chromatid breakage during mitosis (Figure 3). To our knowledge, this is the first study reporting formation of a neocentromere in a segment of chromosome 6p. Our case report demonstrates that mosaic sSMC derived from distal 6p causes severe phenotypes with multi-system malformations, including craniofacial, cardiovascular, genitourinary, ophthalmological, and hearing problems.

Pure partial trisomy 6p is very rare and can be caused by tandem duplications, inverted duplications, a supernumerary marker chromosome, interchromosomal insertions, and unbalanced chromosome rearrangements [11]. The proximal breakpoint in the short arm of chromosome 6 can vary from 6p11 to p25 and the clinical severity varies even within the same family with the same duplicated segments [10]. The typical characteristic features include low birth weight, developmental delay, growth retardation, mild to severe craniofacial abnormalities, feeding difficulties, recurrent infections, neurological anomalies, congenital heart defects, and renal abnormalities [4,10,14]. To our knowledge, only 14 patients with isolated trisomy and one with isolated tetrasomy of 6p with overlapping 6p23-6p25.3 have been reported. Table 3 compares their phenotypes with our case. With partial 6p tetrasomy with a mosaic level of 53%, clinical presentations of our patient were highly analogous to those with trisomy partial 6p, suggesting a gene dosage effect of the contiguous genes within the segment. Stohler et al. proposed that congenital heart defect and renal problems should be considered characteristics of both partial trisomy and partial tetrasomy 6p syndromes [3]. The cardiac defects in distal 6p trisomy are mainly atrial septal defects or persistent ductus arteriosus and the renal problems are characterized with hydronephrosis, proteinuria, and glomerulopathy [12]. Our patient presented as being small for gestational age, having feeding difficulty, developmental delay, and growth retardation. He also had craniofacial dysmorphic features without craniosynostosis, blepharoptosis, or blepharophimosis. There was no seizure nor distinct structure abnormality of the central nervous system. However, a complicated congenital malformation of the cardiovascular and renal systems was also seen in our patient, including complete atrioventricular canal and high-grade vesicoureteral reflux, causing heart failure and recurrent urosepsis, which led to a highly unfavorable outcome. Furthermore, of the 11 cases with reports for hearing function, five (45.5%) had sensorineural hearing impairments to various degrees. Our patient also had bilateral moderate to severe sensorineural hearing impairment, contributive to the poor response to environment. In general, our patient demonstrated the phenotype of mosaic tetrasomy distal 6p was highly consistent with that of trisomy distal 6p.

Our patient had a 14.34-Mb gene dosage increase over the region of 6p23-p25.3 encompassing 65 OMIM genes, mostly not identified as contributory to the phenotype. Of these, two genes, namely *FOXC1* and *BMP6*, might contribute to the phenotype. *FOXC1* (OMIM 601090) is located at 6p25.3 and has been shown to play a role in the regulation of embryonic and ocular development. A recent study suggests that truncating variants of *FOXC1* causes anterior segment dysgenesis and cardiac anomalies [21]. Similarly, in previous reports, eye abnormalities are suggested to be an important component of the phenotype of distal 6p duplication, including congenital cataracts, colobomas, blepharoptosis, and blepharophimosis [10,12]. The dosage effect of *FOXC1* for ocular developmental abnormalities was suggested by a few cases of duplications, involving multiple genes, including *FOXC1,* although this is not well established [22,23,24,25]. Peter’s anomaly, a rare form of anterior segment dysgenesis characterized by central corneal opacity and irido-lenticulo–corneal adhesions, was reported associated with mutations or gene dosage changes of *FOXC1* [22,26]. Furthermore, Jankauskienė et al. suggested that *FOXC1* might play a role in kidney development and postnatal podocyte function [12]. Our patient presented anterior segment dysgenesis diagnosed at 4 days of age, prior to all genetic testing, leading to the impression of Peter’s anomaly. Complicated kidney morphological and functional abnormalities were also presented, including small kidneys, hydronephrosis, hydroureter, vesicoureteral reflux, and proteinuria. This reinforces the hypothesis of triplosensitivity of *FOXC1* which was previously correlated with glaucoma [22,24,25] and anterior segment dysgenesis [23]. On the other hand, overexpression of *BMP6* due to partial trisomy 6p was suggested as being responsible for craniofacial abnormalities including craniosynostosis, choanal atresia, and other dysmorphic features [10]. *BMP6* (OMIM 112266) is located at 6p24.3 and encodes bone morphogenetic protein 6, which has osteogenic activity and involves bone formation [27]. In the literature review, only one out of 12 cases with distal trisomy 6p encompassing *BMP6* presented with craniosynostosis [28], while microcephaly was generally presented in nine patients with an extra dose of *BMP6* gene, except for one family reported by Castiglione et al. [10]. However, of the three patients with distal 6p duplication not encompassing *BMP6*, two presented microcephaly. This result indicates that overexpression of the *BMP6* gene alone does not contribute to microcephaly and craniosynostosis without interplay with other gene components within the region of 6p25.1-p25.3. Consistent with previous cases, our patient presented microcephaly and choanal atresia along with other mild dysmorphic features, but no craniosynostosis.

**Table 3 diagnostics-12-02306-t003:** Detailed clinical features of pure distal tri- and tetrasomy 6p patients reported in the literature.

Reference	Phelan et al. 1986 [29]	Breuning et al. 1977 [5]	Engelen et al. 2001 [6]	Petkovic et al. 2003 [7]	Scott et al. 2007 [8]	Kerrigan et al. 2007 [9]	Castiglioni et al. 2013 [10]	Chen et al. 2014 [11]	Jankauskienė et al. 2016 [12]	Sivasankaran et al. 2017 [4]	Fontana et al. 2017 [13]	Türkyılmaz et al. 2022 [14]	Stohler et al. 2007 [3]	Present Case
Region	6p21-pter	6p21-p25	6p22.1-pter	6p11.1-p25	6p22.2- p25.2	6p25.1-p25.3	6p23-p25.3	6p22.3- p25.3	6p22.1-p25.3	6p22.3-pter	6p25.2-p25.3	6p22.1-p25.3	Tetrasomy 6p25.1-6p25.3.	Mosaic Tetrasomy 6p23-6p25.3.
Age/Sex	NA	2y8m/NA	NA	3y/F	NA/F	8.6y/F	2y/F, 0y/F, 32y/M	Prenatal/F	13y/F	2y/M	32y/F	4m/F	3y/F	0y/M
Low birth weight	+	+	+	+	+	+	−, +, −	+	+	+	NA	+	NA	+
Growth retardation	+	+	-	+	+	+	−, +, −	+	+	+	+	+	+	+
Developmental delay	+	+	+	+	+	+	−, −, −	NA	+	+	-	+	+	+
Microcephaly	+	+	+	+	+	+	-	+	+	+	-	+	+	+
Craniosinostosis	NA	-	-	-	NA	-	-	+	-	-	-	-	-	-
Seizures	NA	-	-	NA	NA	+	-	NA	-	-	-	-	-	
Hemangioma	NA	-	+	NA	NA	+	+, −, −	NA	NA	NA	-	+	NA	+
Craniofacial abnormalities	+	+	+	+	+	+	+, +, −	+	+	+	-	+	+	+
Prominent/ high forehead	+	+	+	+	NA	+	−, −, −	NA	+	-	-	+	+	+
Low-set/ malformed ears	-	+	+	+	NA	NA	+, −, −	+	+	+	-	+	+	+
Bulbous nose	+	+	+	-	NA	+	+, −, −	+	NA	+	-	+	+	+
Choanal atresia	NA	-	-	-	NA	-	+, +, −	NA	-	NA	-	-	-	+
Blepharophimosis/ ptosis	+	NA	+	-	+	+	+, −, −	NA	+	+	-	+	+	-
ACED/Cataracts	NA	-	-	-	-	-	−, −, −	NA	-	NA	-	-	-	+
Hearing loss	NA	NA	-	-	-	-	−, −, +	NA	+	-	-	+	+	+
CHD	-	+	+	-	-	NA	+, −, −	NA	-	-	+	+	+	+
CAKUT	NA	+	+	NA	NA	NA	−, −, −	NA	+	-	-	+	+	+
Small kidney	NA	+	-	NA	NA	NA	−, −, −	NA	+	-	-	-	-	+
VUR	NA	NA	+	NA	NA	NA	−, −, −	NA	-	-	-	-	-	+
Hydronephrosis	NA	NA	+	NA	NA	NA	−, −, −	NA	-	-	-	+	+	+
Proteinuria	NA	+	NA	NA	NA	NA	−, −, −	NA	+	-	-	-	NA	+
Glomerulopathy	NA	+	NA	NA	NA	NA	−, −, −	NA	+	-	-	-	NA	+
Sacral dimple	NA	+	NA	NA	-	NA	−, −, −	NA	NA	-	-	+	+	+
Clinodactyly	NA	NA	NA	NA	-	NA	−, −, −	+	-	+	-	-	+	-
Additions features		Two-lobed right lung; mobile colon ascendens		Mild immunodeficiency (IgG 2/ IgG 4)	Left-sided posterolateral CDH	Hypothalamic hamartoma		Low maternal serum PAPP-A	Corpus callosum hypoplasia, internal hydrocephalus	Macrocephaly			Large AF, bilateral eyelid colobomas, astigmatism hyperopia	

F, female; M, male; y, years; m, months; NA, not available; +, presence; −, absence; ACED, Anterior chamber eye defects; CHD, congenital heart defect; CAKUT, congenital anomalies of the kidney and urinary tract; VUR, vesicoureteral reflux; CDH, congenital diaphragmatic hernia; AF, anterior fontanelle.

## 5. Conclusions

In summary, we presented the molecular cytogenetic characterization of de novo mosaic tetrasomy 6p23-p25.3 because of a neocentric marker chromosome in a male infant, with phenotypes highly analogous to trisomy distal 6p. We reviewed the phenotypes of patients with partial tri- and tetrasomy 6p, and discussed the genotype–phenotype correlation of the involved genes, *FOXC1* and *BMP6*. In patients with microcephaly, growth retardation and malformation of the cardiac and renal system, presentation of the anterior segment dysgenesis might be indicative of a duplication of chromosome 6p. An array-CGH evaluation should be performed for associated syndromic disease.

## Figures and Tables

**Figure 1 diagnostics-12-02306-f001:**
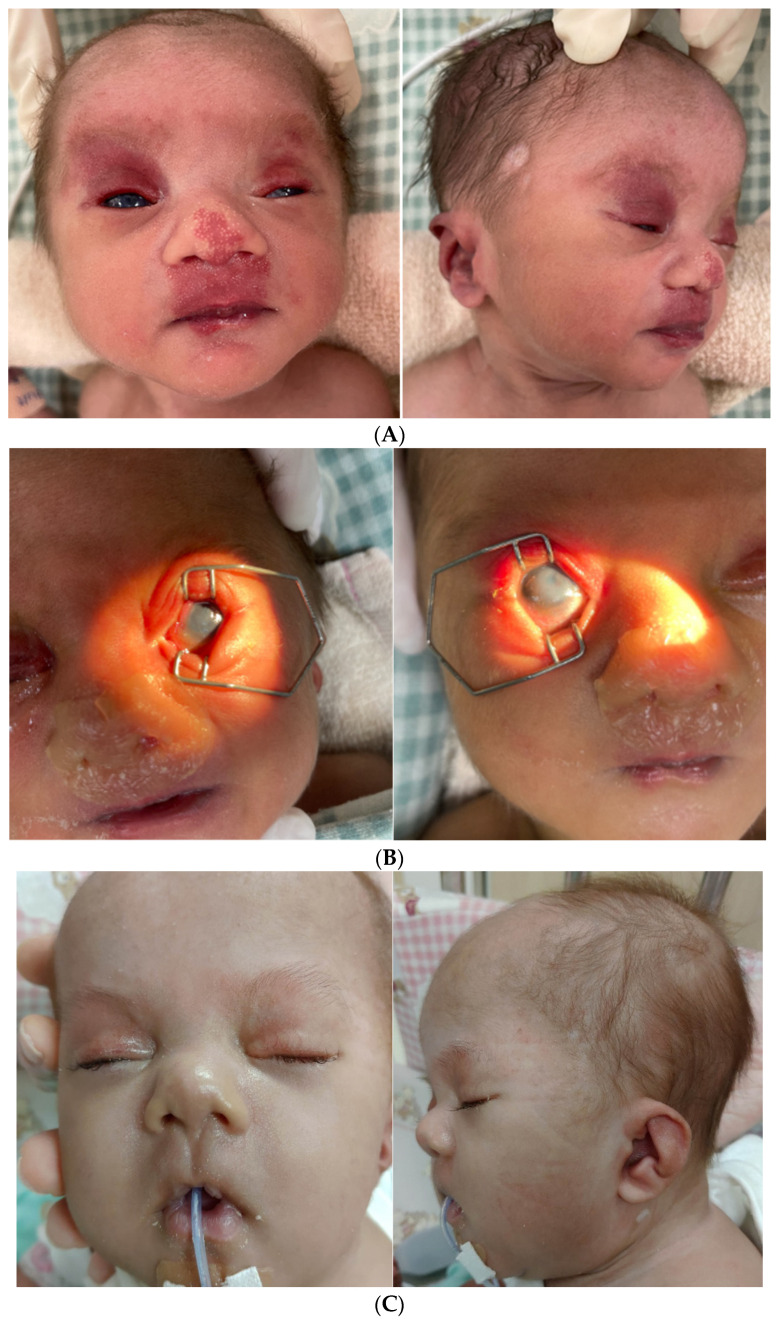
Craniofacial features of our patient. (**A**) At birth, he presented depressed nasal bridge, bulbous nose with prominent columella and bilateral ear lobe crease. Hemangiomas over bilateral upper eyelids, nose tip and philtrum spreading to upper lip. (**B**) At birth, bilateral corneal opacity was also noted. (**C**) The patient’s frontal view and lateral view at 2 months of age. He had depressed nasal bridge, bulbous nose with long philtrum, prominent columella, and left low-set hypoplastic ear.

**Figure 2 diagnostics-12-02306-f002:**
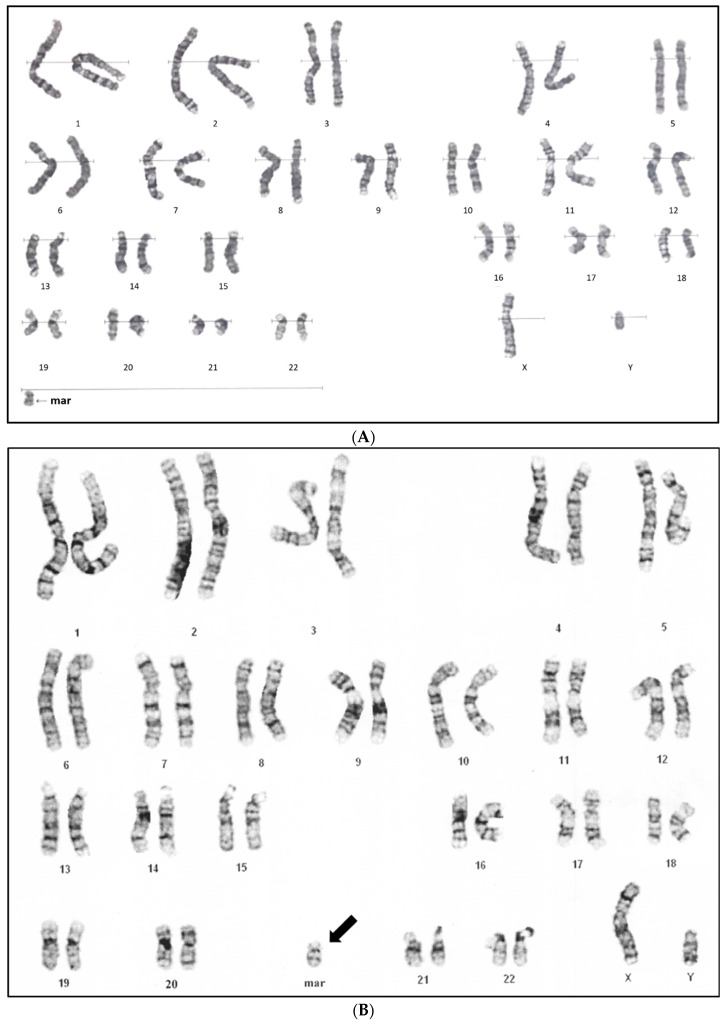
Cytogenetic and molecular genetic study results of our proband. (**A**) Karyotyping of abnormal cells with marker chromosomes at birth. (**B**) Repeated karyotyping of abnormal cells with marker chromosomes at 3 months of age. (**C**,**D**) Array comparative genomic hybridization studies (aCGH) revealed a partial 6p duplication of 14.337Mb at arr[GRCh37] 6p25.3p23 (211198_14548093) × 3–4 with a log2 ratio of 0.598.

**Figure 3 diagnostics-12-02306-f003:**
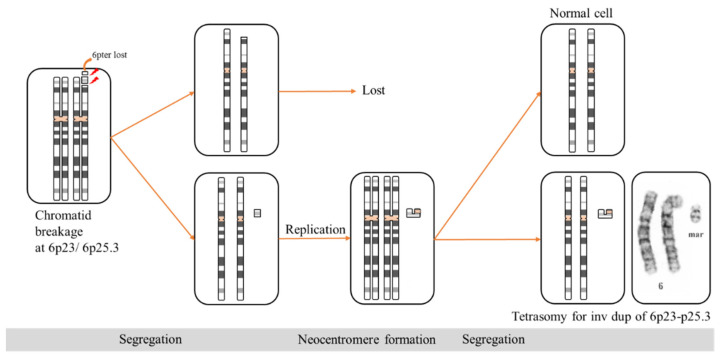
Possible mechanism for the formation of neocentromere after chromatid breakage at mitosis. After chromatid breakage, the acentric fragment of 6p23-p25.3 underwent segregation and subsequent replication. The broken ends of the fragment rejoin to create the inverted duplication, leading to formation of neocentric sSMC. The neocentric sSMC segregates with its sister chromatid, resulting in mosaic tetrasomy for 6p23-p25.3. (Centromeres shown in light orange).

**Table 1 diagnostics-12-02306-t001:** Comparison of cases with clinical findings associated with sSMC(6).

No.	Sex/ Age	Inheritance	Karyotype and Grade of Mosaicism	Result of SMC	Type of Rearrangement	Reference on sSMC Database [17]
1	F/ 1m	dn	47,XX, + r [37]/ 46,XX [13]	r(6)(::p21.2→q10)	Interstitial deletion + r formation	06-W-p21.2/1-1
2	F/ prenatal	dn	47,XY, + mar [12]/ 46,XY [3]	r(6)(::p21.2→q11.2::)	Interstitial deletion + r formation	06-W-p21.2/2-1
3	M/ 14y	NA	47,XY, + mar [12]/ 46,XY [3]	r(6)(p21.2q12)	Interstitial deletion + r formation	06-W-p21.2/2-2
4	M/ 1m	dn	47,XY, + mar [13]/ 46,XY [7]	min(6)(:p21.1→q11:)	Interstitial deletion	06-W-p21.1/1-1
5	M/ 16y	NA	48,XY, + mar, + mar [8]/ 47,XY, + mar [10]/ 46,XY [3]	See note 1 below	Interstitial deletion/Interstitial deletion + r formation	06-W-p12.3/1-1
6	M/ 2y	NA	47,XY, + r [17]/ 48,XY, + rx2[3]	r(6)(p12.3q12)	Interstitial deletion + r formation	06-W-p12.3/2-1
7	M/ 16y	NA	47,XY, + mar [60%]/ 46,XY [40%]	See note 2 below	Interstitial deletion/Interstitial deletion + Inv dup	06-W-p11.2/1-1
8	M/ 2y	dn	48,XY, + marx2[59]/47,XY, + mar [25]/ 46,XY [40]	min(6)(:p11.1→q12:)	Interstitial deletion	06-W-p11.1/1-1
9	F/ 12y	NA	47,XX, + mar [17]/ 46,XX [3]	See note 3 below	Interstitial deletion/Interstitial deletion + r formation/Interstitial deletion + Inv dup	06-W-p11.1/2-1

F, female; M, male; y, years; m, months; NA, not available; dn, de novo; r, ring chromosome; Inv dup, inverted duplication. Note 1: r(6)(::p12.3→q12::)[2]/r(6)(::p12.3→q12::)x2[5]/min(6)(:p12.3→q12:)[9]/min(6)(:p12.3→q12:)x2[2]/min(6)(:p12.3→q1?2::q1?2→p12.3:)[1]/r(6)(::p12.3→q12::),r(6;6)(::p12.3→q12::p12.3→q12::). Note 2: min(6)(:p11.2→q11.1:)[92%]/inv dup(6)(:q11.1→p11.2::p11.2→q11.1:) [8%]. Note 3: min(6)(:p11.1→q13:)[5]/r(6)(::p11.1→q13::)[2]/r(6)(::p11.1→q13::p11.1→q13::)[2]/r(6)(::p11.1→q13::p11.1→q13::p11.1→q13::p11.1→q13::)[1]/ inv dup(6)(:p11.1→q13::p11.1→q13:).

**Table 2 diagnostics-12-02306-t002:** Comparison of cases with clinical findings associated with neocentric sSMC(6).

No	Sex/ Age	Inheritance	Karyotype and Grade of Mosaicism	Result of SMC	Type of Rearrangement	Reference on sSMC Database [17]
1	M/ adult	NA	47,XY,t(4;15), del(6)(q16.2q22.2), + r(6)(::q16.2→q22.2::)[100%]	r(6)(::q16.2→q22.2::)	r formation	06-N-q16.2/1-1
2	M/ 1y	NA	47,XY, + mar [100%]	r(6)(::q24→q25::)	r formation	06-N-q24/1-1
3	NA./ prenatal	dn	47, + mar [60%]/ 46[40%] mar not in fetal blood but in placenta	inv dup(6)(qter→q26: :q26→qter)	Terminal deletion + Inv dup	06-N-qt26/1-1

F, female; M, male; y, years; m, months; NA, not available; dn, de novo; r, ring chromosome; Inv dup, inverted duplication.

## Data Availability

Not applicable.

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
