# Peer review of "De Novo Mosaic 6p23-p25.3 Tetrasomy Caused by a Small Supernumerary Marker Chromosome Presenting Trisomy Distal 6p Phenotype: A Case Report and Literature Review"

_diagnostics, 2022, doi:10.3390/diagnostics12102306_

Round 1
Reviewer 1 Report
The rare case report is presented clearly and has some teaching points. I might suggest to make conclusion shorter.
Author Response
Response to Reviewer 1 Comments
Point 1: The rare case report is presented clearly and has some teaching points. I might suggest to make conclusion shorter.
Response 1: Thank you for your review and suggestion. We would make the conclusion more precise.
Reviewer 2 Report
This is an interesting case report of a patient with 6p23-p25.3 tetrasomy where the authors describe the inverted duplication segment of chromosome 6p, and also compare the patient's clinical phenotype with those with chromosome 6 trisomy previously reported. However this work is clinically and scientifically poorly described with several errors throughout the manuscript, some of them are listed below but I recommend a significant revision of the whole manuscript before publication:
- Abstract: (line 36) wrong karyotype of the male patient is described: 47,XX,+mar[25]/46,XX[22]
- Introduction: several short sentences that should be combined (e.g. The presence of an sSMC causes partial tris- or tetrasomy (line 55); It did not associate with sSMCs (line 59)). Also, what/who is Peter's anomaly? Please refer to FOXC1 and BMP6 genes when referring to the genes that relate to the patient phenotype.
- Results: NG for nasogastric is used only once, please delete it (line 105). urethrocystography should be corrected to cystourethrography (line 116). What do the authors mean by "SMC(6) had inverted duplicated shapes" (line 145)? How did the authors detect this? Why refer FOXC1 and not BMP6 gene (line 147)?
- Discussion: It is good that the possible mechanism of the neocentromeres generation is represented in figure 3, however, the caption should be more clearly described. Explain or reference the triplosensitivity hypothesis of FOXC1 gene (line241) to correlate with the discussion.
- References: review the references since some mistakes were detected. one of them e.g.: authors of reference 1 are wrongly cited, it should be: Jafari-Ghahfarokhi H, Moradi-Chaleshtori M, Liehr T, Hashemzadeh-Chaleshtori M, Teimori H, Ghasemi-Dehkordi P. Small supernumerary marker chromosomes and their correlation with specific syndromes. Adv Biomed Res.
Author Response
Response to reviewer 2 comment
Point 1: Abstract: (line 36) wrong karyotype of the male patient is described: 47,XX,+mar[25]/46,XX[22]
Response 1: Thank you for your comment. We revised the karyotype to 47,XY,+mar[25]/46,XY[22].
Point 2: Introduction: Several short sentences that should be combined (e.g. The presence of an sSMC causes partial tris- or tetrasomy (line 55); It did not associate with sSMCs (line 59)). Also, what/who is Peter's anomaly? Please refer to FOXC1 and BMP6 genes when referring to the genes that relate to the patient phenotype.
Response 2: Thank you for your suggestion. We combined the short sentences in the paragraph. To avoid confusion, we used anterior segment dysgenesis instead of Peter's anomaly in introduction section, and added description and references about Peter's anomaly in discussion section. We also referred to FOXC1 and BMP6 in the last paragraph of introduction section.
Point 3: Results: NG for nasogastric is used only once, please delete it (line 105). urethrocystography should be corrected to cystourethrography (line 116). What do the authors mean by "SMC(6) had inverted duplicated shapes" (line 145)? How did the authors detect this? Why refer FOXC1 and not BMP6 gene (line 147)?
Response 3: Thank you for your suggestion. We revised the sentences accordingly. Besides, we mentioned the inverted duplicated shape mainly to describe the shape of sSMC seen in G-banding. According to T. Liehr (in Benign & Pathological Chromosomal Imbalances, 2014, DOI: 10.1016/B978-0-12-404631-3.00004-0), sSMCs have 3 main shapes, inverted duplicated, centric minute, and ring; and our case presented inverted duplication-shaped sSMC recognized via the banding pattern of sSMC in the karyotype.
Furthermore, thank you again for your suggestion. We referred BMP6 along with FOXC1 in the sentence since it was also discussed later in the article.
Point 4: Discussion: It is good that the possible mechanism of the neocentromeres generation is represented in figure 3, however, the caption should be more clearly described. Explain or reference the triplosensitivity hypothesis of FOXC1 gene (line241) to correlate with the discussion.
Response 4: Thank you for your suggestion. We added figure legend for figure 3 to explain more clearly about the possible mechanism of the neocentromeres generation. We would also explain more about the triplosensitivity hypothesis of FOXC1 and add references reporting possible evidence for triplosensitivity of FOXC1 causing glaucoma, iris hypoplasia [Ref 23, 25-26] and anterior segment dysgenesis [Ref 24].
Point 5: References: review the references since some mistakes were detected. one of them e.g.: authors of reference 1 are wrongly cited, it should be: Jafari-Ghahfarokhi H, Moradi-Chaleshtori M, Liehr T, Hashemzadeh-Chaleshtori M, Teimori H, Ghasemi-Dehkordi P. Small supernumerary marker chromosomes and their correlation with specific syndromes. Adv Biomed Res.
Response 5: Thank you for your suggestion. We reviewed the references, updated all references in the database and revised the references.
